# Multiple Feature Dependency Detection for Deep Learning Technology—Smart Pet Surveillance System Implementation

**Ming-Fong Tsai [1],\*, Pei-Ching Lin [1], Zi-Hao Huang [1]** 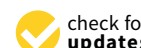 **and Cheng-Hsun Lin [2]**

[1] Department of Electronic Engineering, National United University, Miaoli 36063, Taiwan; a851103123@gmail.com (P.-C.L.); main668888@gmail.com (Z.-H.H.)
[2] Electronic and Optoelectronic System Research Laboratories, Industrial Technology Research Institute, Hsinchu 31040, Taiwan; akiolin@itri.org.tw
\* Correspondence: mingfongtsai@gmail.com

**Abstract:** Image identification, machine learning and deep learning technologies have been applied in various fields. However, the application of image identification currently focuses on object detection and identification in order to determine a single momentary picture. This paper not only proposes multiple feature dependency detection to identify key parts of pets (mouth and tail) but also combines the meaning of the pet's bark (growl and cry) to identify the pet's mood and state. Therefore, it is necessary to consider changes of pet hair and ages. To this end, we add an automatic optimization identification module subsystem to respond to changes of pet hair and ages in real time. After successfully identifying images of featured parts each time, our system captures images of the identified featured parts and stores them as effective samples for subsequent training and improving the identification ability of the system. When the identification result is transmitted to the owner each time, the owner can get the current mood and state of the pet in real time. According to the experimental results, our system can use a faster R-CNN model to improve 27.47%, 68.17% and 26.23% accuracy of traditional image identification in the mood of happy, angry and sad respectively.

**Keywords:** multiple feature; dependency detection; deep learning; surveillance system

## 1. Introduction

In modern society, the population of pets such as cats and dogs is increasing. However, when owners are at work, pets at home will inevitably be alone, and owners might be worried about the safety of pets. Hence, this paper proposes a smart pet surveillance system to automatically identify the pet's mood and state and initiatively send identification results to the owner. In this way, even if owners are busy at work, the pet status will be sent through the smart pet surveillance system. It can quickly grasp the current situation of pets so that owners can work with peace of mind. However, traditional image identification cannot effectively identify the pet's mood and state from a single image or instantaneous state. The pet displays its mood and states through actions of certain barks or continuous movements of several different key parts on its body. Hence, the multiple feature dependency detection algorithm proposed in this paper can be used on most object detection models. The multiple feature dependency detection algorithm can correctly identify the mood and state of the pet when the object detection has sufficient accuracy. Tensorflow architecture of deep learning image identification technology and its Faster-RCNN network architecture extracts a conv-feature map of input images through convolutional layers [1–6]. Then the region propose network (RPN) will process the extracted convolution feature maps and provide a large number of ROIs (region of interest means

regions that may contain feature points). It lets ROIhead (responsible for processing the ROIs proposed by RPN) determine whether there is a feature target in ROIs and correct the position and coordinates of ROIs. Finally, it records specific labels of features (the set featured part). The most important of which is ROIhead, which is responsible for determining whether ROIs contains feature targets and modifying the coordinates and position of ROIs directly affects the accuracy of identification [7–15]. In view of the influence of network architecture design, it is important to improve the identification ability of the identification model, directly change the network architecture and improve the number and quality of input training samples. An effective training sample enables RPN to provide better ROIs so that ROIhead can more accurately detect the target featured parts to be identified to improve the level of identification confidence. The identification system of the smart pet surveillance system proposed in this paper uses deep learning image identification technology, and its pretrained model uses the Faster-RCNN neural network using the COCO (Common Object in Context) data set. In order to identify multi-point of pet features, the proposed system collected training samples of multiple relevant features and trained the identification model. By identifying multiple featured parts of pets and analyzing the continuous changes and relative relationships between featured parts, pet's mood and state are analyzed and determined.

Faster R-CNN can effectively identify more subtle features and then identify the key parts of pets, but it cannot identify the meaning behind images and objects, such as mood and status. Even if the training data of the faster R-CNN can directly label the mood and state, the mood and state cannot be successfully identified, because it cannot be identified only through a single image. Hence, the proposed method in this paper is based on faster R-CNN and extracts the identification results of the key parts of the pet to judge the pet's mood and state. Moreover, the multiple features of the proposed method can be the key parts of the pet itself in the image or the sounds made by the pet. The proposed method uses the KNN (K nearest neighbor) algorithm to establish the speech identification model. The characteristic frequency bands of sound waves emitted by pets in different moods and states are needed to extract and identify and eliminate the noises. The proposed method uses MFCC (Mel-scale frequency cepstral coefficients) to suppress noise in sound waves and extract the sound wave characteristic frequency bands of pets in different moods and states. Since the identification objects are pets of different breeds, colors and sizes, the relative position of the featured part will be different. Even if the breed and color are the same, uncertain factors such as age and hair size will affect the identification ability of the system. In order to enable the identification system to respond to the above-mentioned factors, the system uses the identification system to successfully identify the image of featured parts and recreate a new training sample based on the identification results. By inputting new and customized training samples into the identification system for training, the identification model's ability to identify featured parts of specific pets can be improved. The related literature will be discussed in Section 2, and the system architecture will be explained in Section 3, which will be divided into four small chapters for detailed explanation. In Section 4, we analyze the data recorded by the system during the actual identification process. In Section 5, we present the conclusions of this paper.

## 2. Literature Review

In recent years, there have been many related literatures and applications for image identification, but most of them focused on face identification. Literatures related to animal identification are relatively rare. Related literature [16] focused on wild animal protection, using the image identification system to identify endangered wild animals and protect them. This paper uses convolutional neural networks to train images of endangered wild animals and common animals. The identification system can effectively identify endangered wild animals. However, this paper only identifies the appearance of animals. Limited by its system and training samples, the system can only identify specific creatures, not endangered animals. However, although the system proposed in our paper can only identify common pet species at home, the system can not only identify their appearance but also judge the pet's mood and state by identifying their subtle behaviors. Related literature [17] identified a monkey face

by the identification system trained by a convolutional neural network. Monkeys can communicate through facial expression. It plays an important role in their communication. This paper uses the image identification system to capture the movement of the monkey's facial muscles and its facial expression to determine what the monkey wants to express. This paper uses the movement state of the monkey's facial muscles and changes in the five senses to determine what the monkey wants to express. In the exchange of information, although the face sends a lot of messages, gestures of other body parts can also convey more specific or obvious meaning. The training sample of this paper is sufficient, but it only identifies monkeys' faces. If it includes other physical behavior performance, it should be better at identifying what the monkey wants to express. The system proposed in our paper not only identifies faces but also identifies continuous behaviors of other featured parts including information limb movement can convey. Related literature [18] used transfer learning to design a method for dog identification from shallow to deep. In this paper, different features of dogs such as eyes, noses and ears are identified. Different breeds of dogs are included for identification. Related literature [19] proposes using surveillance systems to monitor the location of pets based on the faster R-CNN identification model. However, our proposed system will identify the mood and state of the pet. Related literature [20] observes the difference in horse behavior before and after surgery, inferring different pain levels based on the condition of the surgical wound and observes the horse's behavioral performance at different levels of pain. This paper only observes the horse's displacement in space. It is concluded that the horse will reduce the displacement when it is in pain. However, our proposed method identifies the dependence of the continuous behavior of different key parts of the pet, instead of identifying the entire pet. Moreover, the system proposed in our paper needs to identify common home pets, using transfer learning to improve the accuracy of identification. It not only identifies the pet's breed, body shape and hair condition to confirm the owner's pet but also the pet's behaviors of featured parts to ensure the pet's mood and state can be correctly identified. Due to individual preferences, the selection of pets is different. In training data collection of the identification system in this paper, the open data set is used and the public pet videos are divided into screenshots and other image data. Therefore, as long as the information of the input training data can be clearly defined, the identification function of the identification system can be improved.

## 3. Materials and Methods

The smart pet surveillance system overview is shown in Figure 1. In order to capture pet videos, the system uses the loop recording subsystem and webcam to capture real-time videos of pets at any time. The loop recording subsystem will continuously capture images and output them as short videos. The recording is not interrupted and the latest short video can be sent to the data processing subsystem. To use image identification and the KNN audio identification model, we need to perform preliminary processing to extract an image and audio signal from the short video. The identification network architecture is shown in Figure 2. This paper was based on the faster R-CNN identification network and customized the key parts of the pet to be identified. Moreover, we used the KNN audio identification model for audio identification. The system automatically processed the images and audio for faster R-CNN and KNN audio identification model to identify the key parts of the pet and the pet's barking. The feature extraction subsystem was used to extract information about the key parts of the pets in these images and the mood contained in the audio. It also generated the training samples for automatic optimization module for supplements when performing automatic training according to the results of the identification of the faster R-CNN identification network. The multiple feature dependency detection algorithm extracted the information from the list of the feature of the pet generated by the feature extraction subsystem. According to the information of these features, we determined the pet's mood and state. Sending the identified pet's message to the owner through the message transmission subsystem so that the owner can understand the pet's latest mood and status. In order to synchronize the identification ability of the identification system with the change of pet's appearance, this system proposed an automatic optimization identification module subsystem that

uses the latest system identification results as samples and sends them to the module for training. It effectively synchronizes the ability of identification model with the changes in the pet's appearance. Functions of the smart pet surveillance system are shown in Figure 3. The proposed system includes an image identification system, which is responsible for identifying the key parts of the pet and the sound of the pet. Multiple feature dependency detection extracts the identified information to determine the pet's mood and status and transmits the information to the owner. Automatic system optimization responds to changes in the pet's appearance.

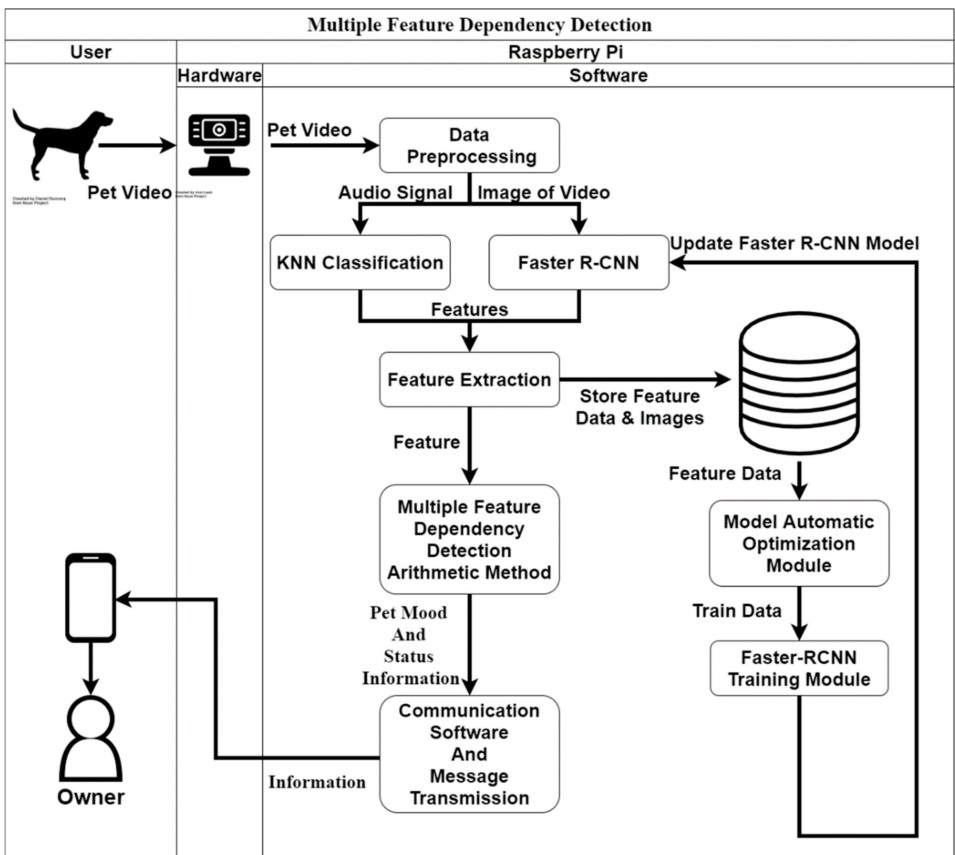

**Figure 1.** Smart pet surveillance system overview.

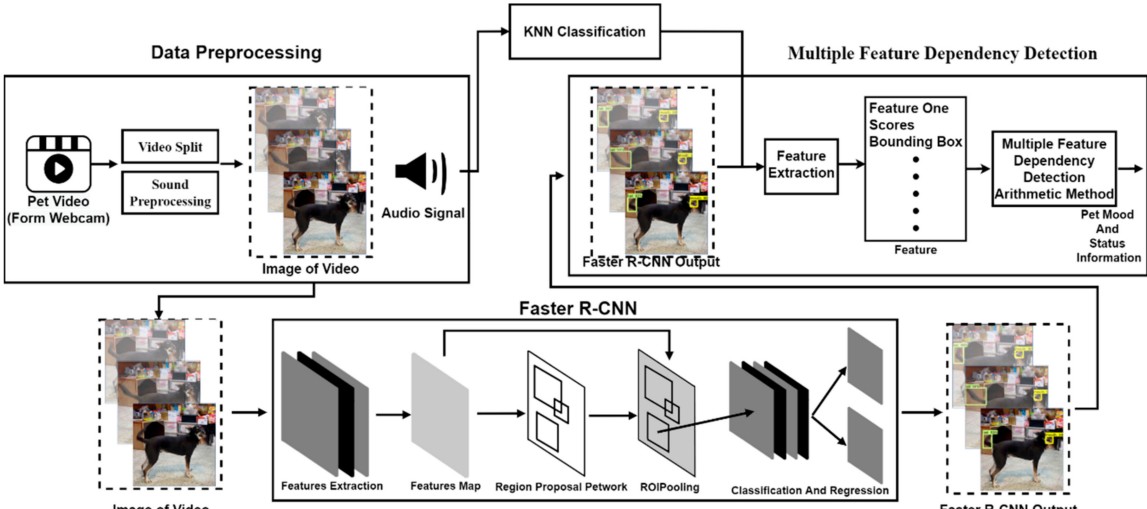

**Figure 2.** Identification network architecture.

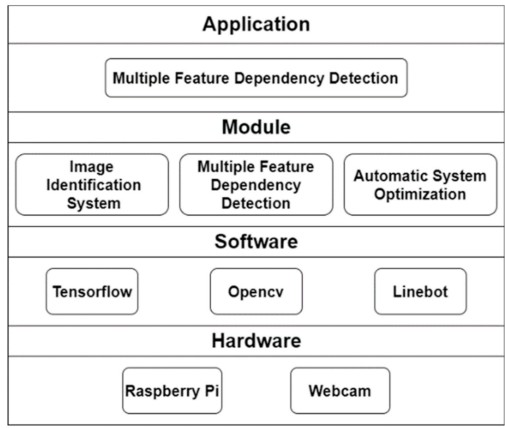

**Figure 3.** Functions of the smart pet surveillance system.

### 3.1. Pet Camera

The smart pet surveillance system proposes to continuously capture real-time images through pet cameras. Since the pet camera and the identification system were implemented in the same operation system, when videos captured by the pet camera were saved in the storage space designated by the identification system, the identification system could use these videos. For convenience of the subsequent image identification system, captured images were saved in short videos when recorded by pet cameras. Therefore, the recorded images need to be continuously stored as shown in Algorithm 1. The loop recording processing recorded the video according to the video format and stored the video, which had a duration setting by time in the video path location.

---

**Algorithm 1** Loop_Recording_Processing

---

**Require:** threading
**Require:** video_store_path *video_path*
**Require:** count variable *time_sup* = 0
1.   **def** timer():
2.      *time_sup* += 1
3.   **main**():
4.      **threading.**timer()
5.      set video format and information: *v_ format*
6.      **while**(1):
7.         set video duration: *time*
8.         **if** *time_sup* **==** *time*:
9.            store the video to *video_path*
10.           *time_sup* = 0
11.        **else:**
12.           recording the video according to *v_ format*
13.     **return 0**

---

### 3.2. Identification System

As shown in Algorithm 2, the identification subsystem of the smart pet surveillance system was implemented by object identification. By detecting and identifying the key part of the pet, it is able to identify the different pet's mood and state by changing different key parts of the pet. When the identification subsystem is activated, it will automatically read the short videos recorded and stored by the pet camera. Owing to the identification method that uses faster R-CNN and the KNN audio identification model, the video needs to extract image and audio files, then these are sent to the respectively identification system for identification. After the video is divided into images, it will have

different results according to different sampling parameters. When the sampling parameters are set too low, time interval between two images will be too long, resulting in few identifiable images to effectively identify the mood and state of pets. When the sampling parameters are set too high, there will be too many segmented images, resulting in excessive load of the system and greatly increasing identification time. Therefore, in this paper, videos were cut to five to six images per second for identification. After the data processing subsystem red the video, it separated and stored the image and feature of the audio according to the sampling parameters set by the system. The system processed through MFCC and extracted the feature parts in the audio. When the convert process ended, the system threw the images extracted from the video and feature of the audio into the faster R-CNN identification system and KNN audio identification system for identification.

---

**Algorithm 2** Data_Preprocessing

---

**Require:** Loop_Recording_Processing_output *video_path*
**Require:** image_save_path *i_path*
**Require:** audio save_path a_*path*
1.   **def** video_to_img(*video_path, i_path*):
2.       images obtained from the video: *i_image*
3.       i_image store to i_path
4.   **def** video_to_wav(*video_path, v_path*):
5.       *audio* = audio from the video
6.       *mfcc_audio* = audio use mfcc for feature extraction
7.       mfcc_*audio* store to a_*path*
8.   **main**():
9.       video_to_img(*video_path, i_path*)
10.      video_to_wav(*video_path*, a_*path*)
11.      **return** 0

---

As shown in Algorithm 3, the feature extraction subsystem will extract all the key parts of pets from the identification results of faster R-CNN information in the feature list. Additionally, it will then extract the sounds identification result from the KNN audio identification model. The identification results of the faster R-CNN and KNN audio identification system were used to generate feature lists with feature categories for subsequent processing. Identifying the mood and state condition of pets requires observation of the continuous changes of their characteristics. This is a natural phenomenon of time continuity and object dependence, so it is impossible to determine the pet's mood and state via a single identification result. It is necessary to continuously identify that the featured parts meet specific conditions to determine that the pet conforms to a specific mood or state. It is also necessary to filter information in identification results and exclude images in specific situation where no featured parts are detected to improve the identification effectiveness of the system. The multiple feature dependency detection algorithm is shown in Algorithm 4. The multiple feature dependency detection algorithm contains a number of emotions, which was identified through continuous image and sound features. The multiple feature dependency detection algorithm obtains a feature list via feature extraction and extracts features needed to identify emotions from the feature list according to different emotion requirements. The proposed algorithm determines whether pets are in a certain emotion state according to the nature of the continuity of emotions. When there are insufficient features to make judgments, the proposed algorithm defines pets as in the normal state. For example, after the identification system receives images, it automatically identifies whether there are specified featured parts in the image, such as the dog's mouth keeps opening and tail shaking. After capturing multiple specific features, the system will record the above information. Then, it judges if the above-mentioned image is a continuous one. Therefore, discontinuous images without time continuity cannot represent the mood and state of

pets. The proposed system that obtains the above-required information can identify the pet's mood and state.

---

**Algorithm 3** Feature_Extraction

---

**Require:** Faster R-CNN output list *faster_output_list*
**Require:** KNN audio model output *knn_output*
**Require:** path of all unprocessed training data *data_path*
1.   a dict of feature information: *feature_dict*
2.   **for** *feature_info* **in** *faster_output_list*:
3.     *feature_dict* append feature information from *feature_info*
4.   *feature_dict* append *knn_output*
5.     store feature data & image to *data_path*
6.   **return** *feature_dict*

---

**Algorithm 4** Multiple_Feature_Dependency_Detection_Arithmetic_Method

---

**Require:** Feature_Extraction output *feature_dict*
1.   **def** mood(*feature_dict*):
2.     from *feature_dict* get information to judge pet mood: *data*
3.     **if** *data* is enough to judge mood:
4.       **return** "mood"
5.     **else**:
6.       **return** "normal"
7.   **main**():
8.     list of mood: *mood_list*
9.     list of pet status: *pet_status*
10.    **for** mood **in** *mood_list*:
11.      *pet_status* = mood(*feature_dict*)
12.      **return** *pet_status*

---

*3.3. Communication Software and Message Transmission*

The system uses the multiple feature dependency detection subsystem to transform images and audio into information, which includes the mood and state of pets. It uses communication software with a high penetration rate allowing information identified by the system to be directly transmitted to the application on the owner's smart phone. As shown in Algorithm 5, after setting the information required by the communication software, this subsystem will set different messages according to different emotions identified in the multiple feature dependency detection algorithm and send to users' communication software so that users obtain the latest information about their pets.

---

**Algorithm 5** Message Transmission

---

**Require:** pet_status *status*
1.   owner_id: *id*
2.   dict of message: *dict_msg*
3.   **for** msg_mood **in** dict_msg.keys():
4.     **if** msg_mood == *status:*
5.       send pet's status message to the *id*

---

*3.4. Automatic System Optimization*

The smart pet surveillance system proposes to identify the pet's mood and state by identifying the dependencies of multiple featured parts of pets. The above-mentioned specific featured parts will vary with age of the pet and seasons. The model used by the identification subsystem must be updated with

time. Multi-featured parts dependency identification model in this subsystem needs to be continuously optimized. The identification model synchronizes with the changes of pets so that the system can get the most accurate identification results. Since the image identification system uses deep learning architecture, sufficient training samples are required in order to send data to the identification model for training. The acquisition, labeling and training of the above-mentioned samples are functions of this subsystem. Core functions are shown in Algorithm 6. This subsystem uses images used in previous identification and identification results to extract featured parts of original images. In addition to the image itself, the training samples also need to label the featured parts contained in this image. Figure 4 shows an image that has been identified, this picture contains two features that the system needs to determine the pet's mood and state, namely the opened mouth and tail, and thus, it can be extracted based on the above results to form a new training sample. The system will mark it with the relevant information obtained during identification, the image and its contained information are completely saved, and the subsequent generated identification model is more complete.

---

**Algorithm 6** Model_Automatic_Optimization_Module

---

**Require:** path of all unprocessed training data from Feature_Extraction: *data_path*
**Require:** path of all processed training data *t_path*
1.  **def Retraining_Data_Generation_Module**(*data_path*):
2.     get data and images stored for training: *data_list*
3.     **for** data **in** *data_list***:**
4.        *image* = cut the original image according to x, y
5.        labeling *image* base on *x, y* coordinates and store to *t_path*
6.  **main**():
7.     Retraining_Data_Generation_Module(*data_path*)
8.     Faster R-CNN Training Module(*t_path*)

---

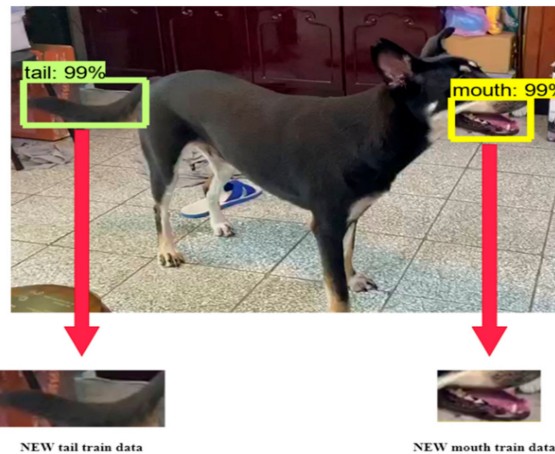

**Figure 4.** Retraining data samples.

## 4. Results and Discussion

### 4.1. System Execution Result

In order to evaluate the practicability and reliability of the proposed system, this paper used an actual pet video and invited testers to verify whether the model correctly identified the mood and state of pets. The environment of the proposed system used Tensorflow-gpu-1.14.0, and the pretraining model used the Faster-RCNN-Inception-V2. The system implementation is shown in Figure 5. The proposed system used training samples to retrain the model for capturing specific featured parts. The training samples included the Stanford Dogs Dataset [21], which contains at least

40 breeds and more than 500 dogs. As shown in Figure 6, there were various pets in training samples to enhance identification ability of faster R-CNN, ssd mobilenet and yolo identification models and avoid over-fitting. The total training images were 2761 and training step was 50,000. In addition to the established Tensorflow-gpu-1.14.0 environment, Raspberry Pi was used by the pet camera with the main system, all subsystems and related modules as shown in Figure 7. As shown in Algorithm 7, the main system will automatically execute the related subsystems for image recording, file conversion, image identification, mood and status determination and other related subsystems.

---

**Algorithm 7** main

---

**Require:** threading
**Require:** Loop_Recording_Processing
**Require:** Data_Preprocessing
**Require:** Faster R-CNN
**Require:** KNN audio model
**Require:** Feature_Extraction
**Require:** Multiple_Feature_Dependency_Detection_Arithmetic_Method
**Require:** Message Transmission
**Require:** Model_Automatic_Optimization_Module
1.　**def** recording:
2.　　**threading.**Loop_Recording_Processing()
3.　**def** Model_Automatic_Optimization_Module:
4.　　**threading.**Model_Automatic_Optimization_Module()
5.　**main**():
6.　　recording()
7.　　**while**(1):
8.　　　Data_Preprocessing()
9.　　　*feature_dict* = Feature_Extraction(Faster R-CNN(), KNN audio model())
10.　　　*pet_status* = Multiple_Feature_Dependency_Detection_Arithmetic_Method(*feature_dict*)
11.　　　Message Transmission(*pet_status*)

---

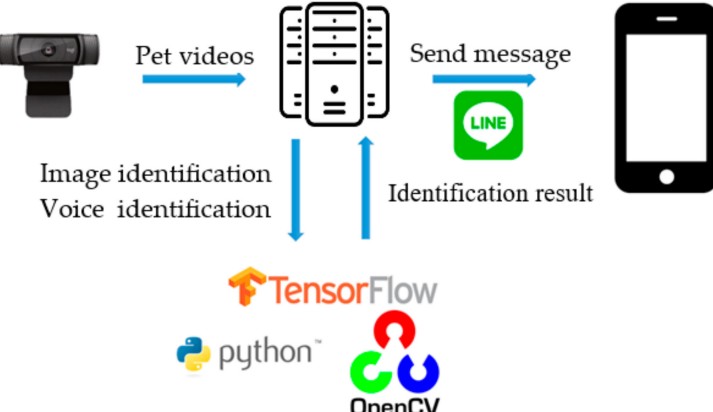

**Figure 5.** System implementation.

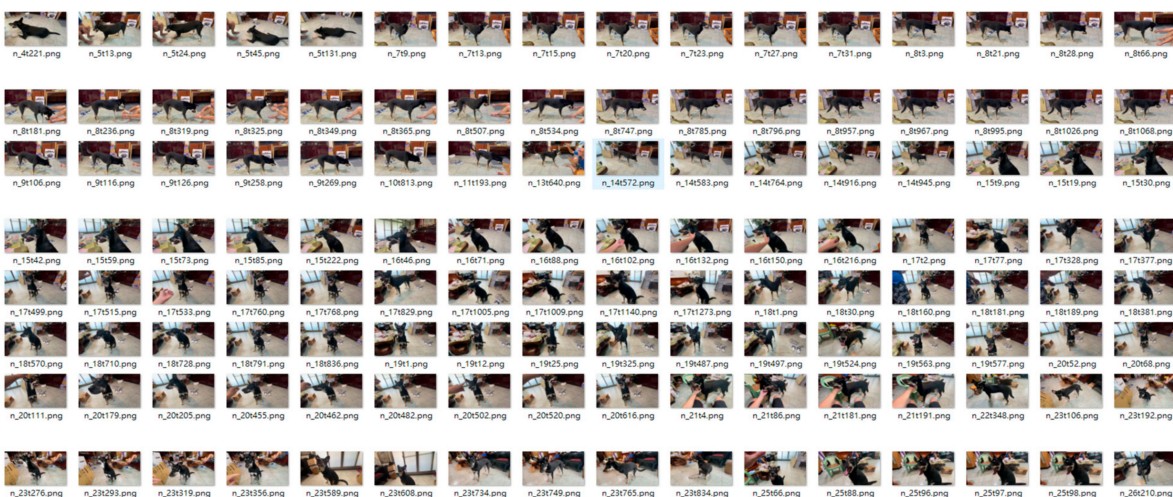

**Figure 6.** Training samples.

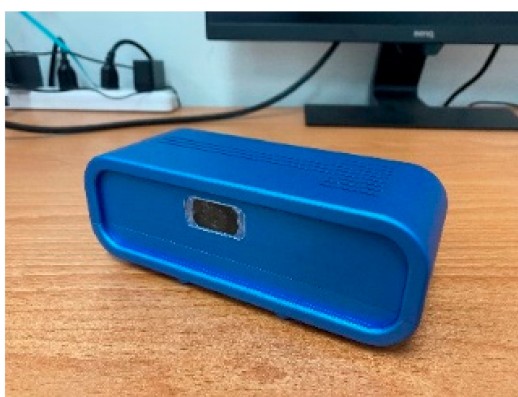

**Figure 7.** Pet camera.

A loop recording subsystem was set in the server to control the webcam and the Tensorflow environment was used for identification and training. The server also stored all videos captured by the webcam. The training sample file generated by the system program was used for subsequent automatic optimization for the identification model. The system continuously captured pet images through the front-end webcam and then saved the image to the path specified by the identification environment. After the image of the pet was segmented and processed, an image to be identified was obtained. The above-mentioned image was identified and the identified featured parts on the image were recorded. In order to verify the effectiveness of mood and state identification of the smart pet surveillance system, we selected to judge happy and angry mood of dogs. In order to reduce the computational burden of the identification system, this paper set the confidence level as 90%. When the confidence level of the object detection was lower than 90%, subsequent algorithms would directly give up this identification result. After the images and audio images extracted from the short movie were identified by faster R-CNN and the features extracted using the feature extraction subsystem, the pet mood and status identified by the multiple feature dependency detection algorithm would send the message to the owner's smartphone through the message transmission subsystem. After the information was filtered and the state was identified, the information would be transmitted to users through the Internet as shown in Figure 8.

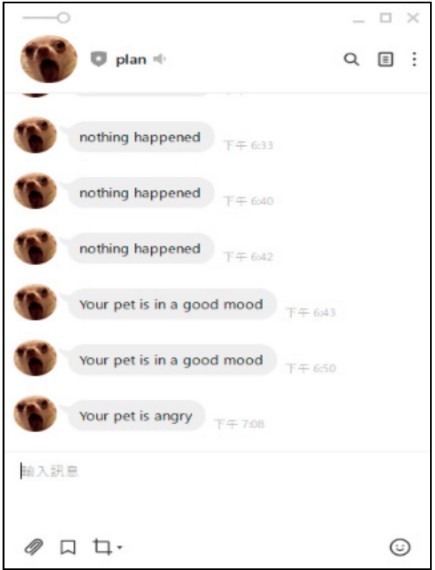

**Figure 8.** Communication software.

*4.2. Types of Mood and Ways of Judging*

The happy characteristics were a continuous mouth open and swinging tail. In order to confirm whether there was a dog in the picture, we needed to mark and identify the dog. To identify the dog's mood and state it as happy, the two marked and identified key parts of the dog were the dog's mouth and the dog's tail. The dog's mouth needed to open continuously and the tail needs to swing in a significant angle. The multiple feature dependency detection algorithm needed to extract the classes of the dog, dog's mouth and dog's tail from the feature list as the basis for determining happy emotions. The system first captures the position of the dog in the image and then identifies whether the dog's mouth is open. When the dog's mouth is open, the system will capture the coordinate value of the dog's tail. Then, we stored the image number and coordinate value to determine whether the number of the image in the list was continuous and the change in the coordinate value of the tail. The identify results of the multiple feature dependency detection algorithm are shown in Figure 9. The multiple feature dependency detection algorithm can effectively use the key parts of the pet to identify the pet's mood and state. The different barking changes made by pets can directly express different moods and states. When we cannot identify the mood and state of the pet through the image alone, it will temporarily determine that the pet's mood is normal, and then perform mood detection with image features and sound features. The mood can be identified as normal, angry and sad when images and sounds are used as features for identification. Since sound is added as a feature, the image will be identified whether there is a change in the state of the dog's mouth to determine that the barking is made by the dog. The multiple feature dependency detection algorithm will extract the dog and the dog's mouth open and close state from the feature list. The class identification based on the above three images was used to determine whether the barking sound was made by the dog in the screen. Moreover, the mood represented behind the barking identified using the KNN audio identification model must be extracted. When the dog barks, the dog's mouth will open and close alternately. As shown in Figure 10, the system first captured the position of the dog in the image, and then identified the state of the dog's mouth, and stored the state of the dog's mouth in the list according to the image number. When the state of the dog's mouth in the list continuously opens and closes, the algorithm will determine the mood and state of the pet based on the audio identification result. Since the pet displays its mood and states through actions of certain barks or continuous movements of several different key parts on its body, we designed the rules of multiple feature dependency detection for the pet's mood and states in Table 1.

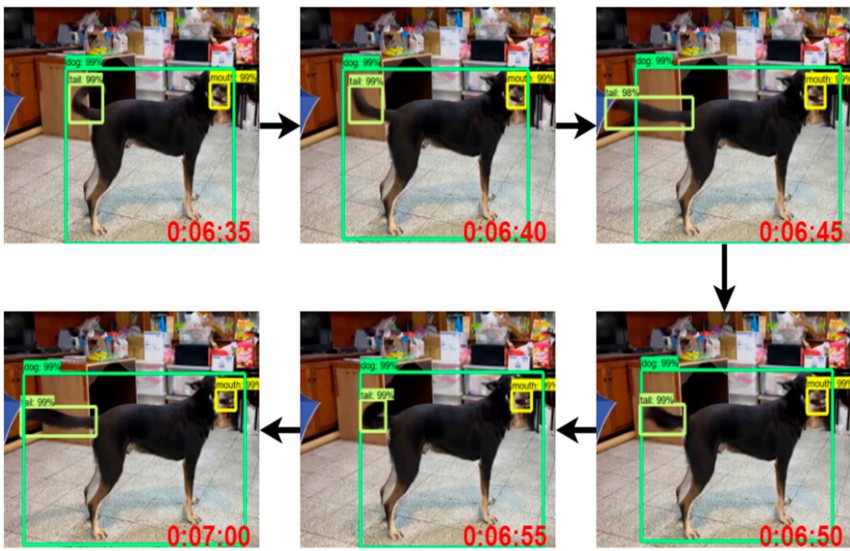

**Figure 9.** Identify mood and state with images.

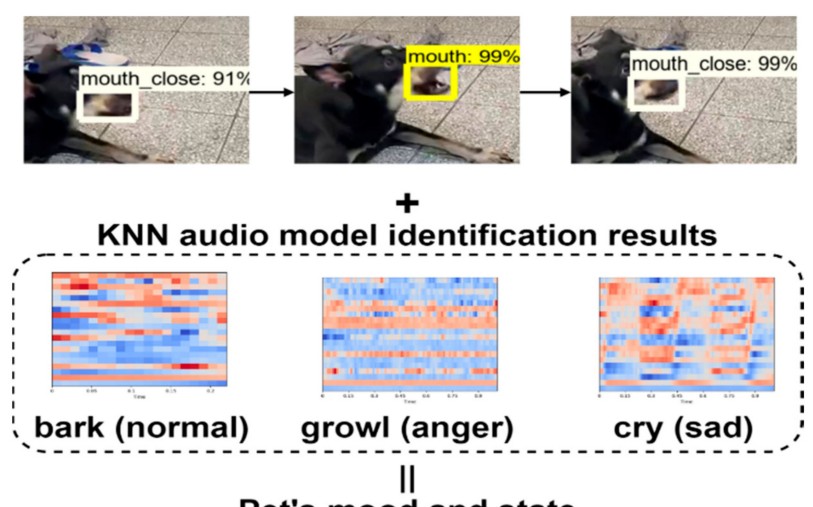

**Figure 10.** Continuous image combined with sound identification results.

**Table 1.** Rules of multiple feature dependency detection.

| Feature / Mood | Image | Voice |
|---|---|---|
| Happy | mouth keeps opening and tail swing continuously | Not needs |
| Angry | mouth keeps opening and closing | Growl |
| Sad | mouth keeps closing | Crying |
| Normal | other actions cannot be identified as any of the above moods | |

## 4.3. The Accuracy of the Model for Identifying Features

As shown in Figure 11, we tested the identification accuracy of each model in identifying the key parts of the pet and combined the identification results of key parts with multiple feature dependency detection algorithm to determine the pet's mood and state results. Faster R-CNN had the higher identification accuracy for key parts of pets, under an average accuracy of 80.95%, than ssd mobilenet v2 at least 55% identification accuracy. Since the proposed method was based on the accuracy of identified key parts of the pet, the accuracy of faster R-CNN obtained 61.1% accuracy of happy category. The reason for low accuracy of the happy state was that one of two features could not be identified by

the faster R-CNN in the continuous image. Theoretically, the identification accuracy of the pet's feature parts improving that multiple feature dependency detection algorithm will more accurately identify the pet's mood and state, such as yolo v3. Hence, the proposed method of this paper could choose fast R-CNN or yolo v3 to identify key parts of pets. This paper tested the identification accuracy of directly using emotions as labels. A faster R-CNN model could reach 33.63% accuracy in identifying the emotions, but the yolo v3 model could reach 55.75% accuracy in identifying the emotions.

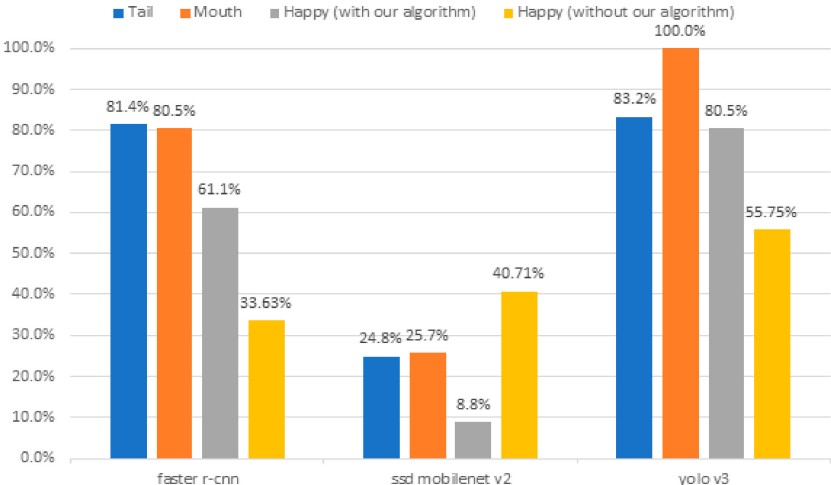

**Figure 11.** Identification of the accuracy for determining the happy state.

This paper tested the accuracy of the KNN audio identification model where the K value (k nearest neighbors) as 3 was used to identify the mood represented by the dog's voice. Moreover, we tested the impact of the different number of features extracted from each training data on the accuracy of the KNN audio identification model. There were 111 audio data. The time of each audio data was one second, including 37 barking (representing normal), 37 growling (representing anger) and 37 crying (representing sadness) [22]. The number of features extracted from each audio data for the KNN audio identification model training was 20, 30 and 40, respectively. We used the leave-one-out cross validation to test the accuracy of the KNN audio model. As shown in Table 2, we tested the impact of MFCC preprocessing on the accuracy of the KNN audio identification model and the accuracy of each audio training data extracting different amounts of data as features. The average accuracy was 60.06% when the audio training data was not preprocessed with MFCC. The average accuracy of the KNN audio identification model generated by training data was 80.48% when the audio training samples using MFCC for preprocessing were used. In Figure 12, we tested not only the identification accuracy of each model in directly identifying the mood of the pet but also the identification accuracy of our method to determine the effectiveness of our system. Without our method, a yolo v3 model averagely reached 19.78% accuracy. Conversely, our system averagely reached 55.81% accuracy in comparison to a yolo v3 model. Without our method, a faster R-CNN model averagely reached 22.85% accuracy. Conversely, our system averagely reached 70.05% accuracy in comparison to a faster R-CNN model. Our system could identify the mood and state of the pet more accurately when the accuracy of image and sound identification is improved.

**Table 2.** The accuracy of the K nearest neighbor (KNN) audio identification model.

| Feature Number | 20 (without/with) MFCC | | | 30 (without/with) MFCC | | | 40 (without/with) MFCC | | |
|---|---|---|---|---|---|---|---|---|---|
| | Barking | Growl | Crying | Barking | Growl | Crying | Barking | Growl | Crying |
| Bark | 33/34 | 8/1 | 9/14 | 35/35 | 6/1 | 6/13 | 32/35 | 5/1 | 3/13 |
| Growl | 2/0 | 19/33 | 16/2 | 1/0 | 18/33 | 17/2 | 3/0 | 21/33 | 18/2 |
| Cry | 2/3 | 10/3 | 12/21 | 1/2 | 13/3 | 14/22 | 2/2 | 11/3 | 16/22 |
| Average | 57.66%/79.28% | | | 60.36%/81.08% | | | 62.16%/81.08% | | |
| Accuracy | 60.06%/80.48% | | | | | | | | |

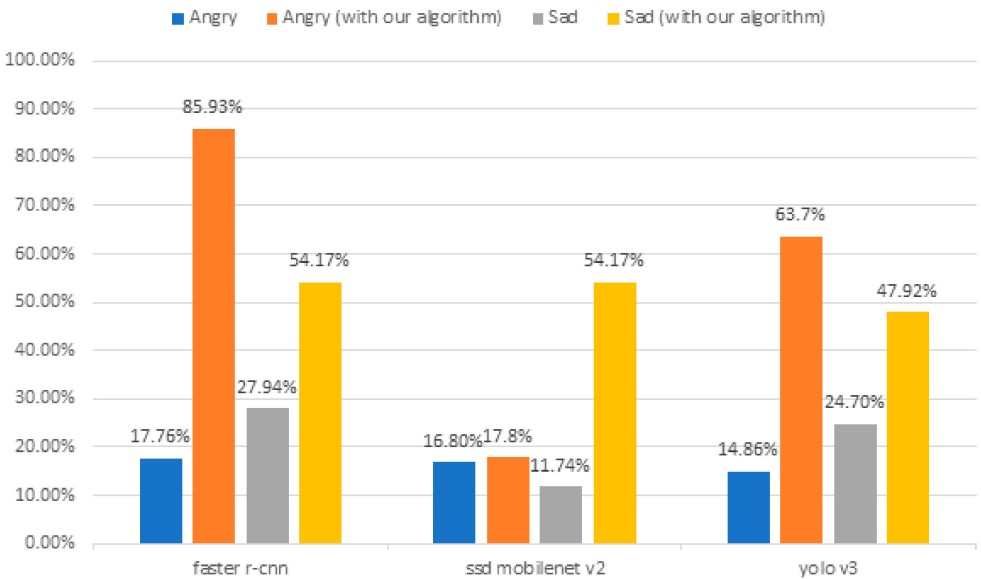

**Figure 12.** Identification of the accuracy for determining angry and sad states.

## 5. Conclusions

This paper proposed multiple featured parts and a time-continuous state of featured parts to identify the pet's mood and state. Additionally, the feature was not only the image, but the sound could also be one of the features. Extending image identification from the identification of a single image had time continuity and multiple object dependencies, which greatly increased the scalability of image identification technology applications. The proposed system used images of specific featured parts that were successfully identified and sent feedback to the system's automatic optimization subsystem to achieve model identification capabilities that kept up with the times. The proposed image identification module, KNN audio identification model and the multiple feature dependency detection algorithm were checked at various levels to make sure core functions of the smart pet surveillance system were feasible. Therefore, as long as the mood and state can be defined by featured parts of pets, the system can immediately identify other moods or states. With the automatic optimization subsystem included in the system, it is able to improve the identification ability. The smart pet surveillance system proves that the image identification with time continuity can be applied in a wider range than the traditional single image identification.

**Author Contributions:** M.-F.T. developed the main idea, supervision, writing—original draft and writing–review & editing; P.-C.L. and Z.-H.H. developed the writing—original draft, and C.-H.L. developed project administration. All authors have read and agreed to the published version of the manuscript.

**Funding:** This research was funded by Industrial Technology Research Institute, Taiwan and the APC was funded by National United University, Taiwan.

**Conflicts of Interest:** The authors declare no conflict of interest.

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
