# Peer review of "Multiple Feature Dependency Detection for Deep Learning Technology—Smart Pet Surveillance System Implementation"

_electronics, doi:10.3390/electronics9091387_

Round 1
Reviewer 1 Report
The paper describes the overall design of a system for the automatic assessment of pet well-being and communication to owners.
Background: the authors miss to discuss several works on the assessment of pain in farm animal (horses, sheep), which is close to the pursued here.
System: the authors only describe the overall system, without enough details about it. There is no discussion of user requirements, technological options and support for the decisions made at the design.
Automatic Recognition of pets and well-being: again, it is even not clear exactly what is being classified, in how many classes, etc.
It is not clear the evaluation protocol and the difficulty of the task, etc.
Summing up, it seems it is a (probably) goof engineering project but without sufficient scientific relevance to support the acceptance of the paper.
Author Response
Revision Summary
Paper Title: Multiple Feature Dependency Detection for Deep Learning Technology – Smart Pet Surveillance System Implementation
Dear Editors and Reviewers,
We would like to express our greatest gratitude for you, the editorial team, and reviewers who have given us valuable comments to improve this paper. We have addressed all comments in the revision and listed the modifications as follows.
Ming-Fong Tsai
Department of Electronic Engineering
National United University
Taiwan
________________________________________________________________
Response to Reviewer 1’s Comments
The paper describes the overall design of a system for the automatic assessment of pet well-being and communication to owners.
Reviewer’s comment 1:
The authors miss to discuss several works on the assessment of pain in farm animal (horses, sheep), which is close to the pursued here.
Author’s response:
We thank the reviewer for the Reviewer 1‘s comment about our paper. We have added the related works in Section 2. This edit is shown in the revised version. Please refer to Section 2.
Reviewer’s comment 2:
The authors only describe the overall system, without enough details about it. There is no discussion of user requirements, technological options and support for the decisions made at the design.
Author’s response:
We thank the reviewer for the kind reminder. We have included more detailed explanation of overall system. This edit is presented as in the revised version. Please refer to Section 1 and Section 3.
Reviewer’s comment 3:
Automatic Recognition of pets and well-being: again, it is even not clear exactly what is being classified, in how many classes, etc.
Author’s response:
We thank the reviewer for the valuable comments. We have included more detailed explanation of automatic recognition. This edit is presented as in the revised version. Please refer to Section 3 and Section 4.
Reviewer’s comment 4:
It is not clear the evaluation protocol and the difficulty of the task, etc.
Author’s response:
We thank the reviewer for the kind reminder. The revised version has been modified according to the reviewer’s comment. Please refer to Section 4.
Reviewer’s comment 5:
Summing up, it seems it is a (probably) good engineering project but without sufficient scientific relevance to support the acceptance of the paper.
Author’s response:
We thank the reviewer for the Reviewer 1‘s comment about our paper. We have added comparisons between our research and similar studies in the Section 4, thereby highlighting the performance of our proposed method. Please refer to Section 4.

Reviewer 2 Report
The revised manuscript is well written. However, authors should sophisticate the manuscript in detail.
Major:
The manuscript has ignored the instructions for authors. (https://www.mdpi.com/journal/electronics/instructions)
The sections are needed to create according to the guideline in “Research Manuscript Sections”.
There is no description of the limitation of this study. Please add the limitation of this study. Especially in recent years, the object detection techniques with deep learning have been improved using various CNN; YOLO, SSD, and so on. Please refer to some related manuscripts to compare with this study.
There are no numerical results on the abstract. The abstract of scientific papers should be written the numerical result, which brought to a conclusion.
There is no information about the dataset. The authors should show detailed information about the number of training and test data.
Minor:
There is no description of the training environments for creating models of your proposed method; programming language, machine specification for training, software, and so on. Please add detailed information.
How many testers were assign to this study? The authors should add detailed information about testers.
Author Response
Revision Summary
Paper Title: Multiple Feature Dependency Detection for Deep Learning Technology – Smart Pet Surveillance System Implementation
Dear Editors and Reviewers,
We would like to express our greatest gratitude for you, the editorial team, and reviewers who have given us valuable comments to improve this paper. We have addressed all comments in the revision and listed the modifications as follows.
Ming-Fong Tsai
Department of Electronic Engineering
National United University
Taiwan
________________________________________________________________
Response to Reviewer 2’s Comments
The revised manuscript is well written. However, authors should sophisticate the manuscript in detail.
Reviewer’s comment 1:
The manuscript has ignored the instructions for authors. The sections are needed to create according to the guideline in “Research Manuscript Sections”.
Author’s response:
We thank the reviewer for the kind reminder. The revised version has been modified according to the reviewer’s comment.
Reviewer’s comment 2:
There is no description of the limitation of this study. Please add the limitation of this study. Especially in recent years, the object detection techniques with deep learning have been improved using various CNN; YOLO, SSD, and so on. Please refer to some related manuscripts to compare with this study.
Author’s response:
We thank the reviewer for the Reviewer 2‘s comment about our paper. The revised version has been modified according to the reviewer’s comment. This edit is presented as in the revised version. Please refer to Section 4.
Reviewer’s comment 3:
There are no numerical results on the abstract. The abstract of scientific papers should be written the numerical result, which brought to a conclusion.
Author’s response:
We thank the reviewer for the Reviewer 2‘s comment about our paper. The revised version has been modified according to the reviewer’s comment. Please refer to Abstract.
Reviewer’s comment 4:
There is no information about the dataset. The authors should show detailed information about the number of training and test data.
Author’s response:
We thank the reviewer for the Reviewer 2‘s comment about our paper. The revised version has been modified according to the reviewer’s comment. This edit is presented as in the revised version. Please refer to Section 4.
Reviewer’s comment 5:
There is no description of the training environments for creating models of your proposed method; programming language, machine specification for training, software, and so on. Please add detailed information. How many testers were assign to this study? The authors should add detailed information about testers.
Author’s response:
We thank the reviewer for the Reviewer 2‘s comment about our paper. The revised version has been modified according to the reviewer’s comment. This edit is presented as in the revised version. Please refer to Section 4.

Round 2
Reviewer 1 Report
Although I acknowledge the effort made to improve the document, the work still presents the same limitations I identified before, and the paper itself still lacks quality:
-some of the pseudo-code does not add anything to the textual presentation
-it is difficult to assess the quality of the detection task and mood classification: the reported results are not detailed enough.
-It is not clear the evaluation protocol and the difficulty of the task, etc. How many different dogs (individuals) were used? How many different dog breeds were used? Were the photos acquired in the wild, etc?
-Is kNN suitable for real-time operation? The authors suggest new data will be used to update models, which in kNN could mean increasing the 'database of observations' and therefore increase inference time.
-" The total training images are 458 "? is this enough?
-Was data augmentation used?
-etc.
Author Response
Revision Summary
Paper Title: Multiple Feature Dependency Detection for Deep Learning Technology – Smart Pet Surveillance System Implementation
Dear Editors and Reviewers,
We would like to express our greatest gratitude for you, the editorial team, and reviewers who have given us valuable comments to improve this paper. We have addressed all comments in the revision and listed the modifications as follows.
Ming-Fong Tsai
Department of Electronic Engineering, National United University, Taiwan
________________________________________________________________
Response to Reviewer 1’s Comments
Although I acknowledge the effort made to improve the document, the work still presents the same limitations I identified before, and the paper itself still lacks quality.
Reviewer’s comment 1:
Some of the pseudo-code does not add anything to the textual presentation.
Author’s response:
We thank the reviewer for the Reviewer 1‘s comment about our paper. We have been make sure all of pseudo-code are added the textual presentation. Moreover, we rewrite the pseudo-code for easy to understand. Please refer to Section 3 and Section 4.
Reviewer’s comment 2:
It is difficult to assess the quality of the detection task and mood classification: the reported results are not detailed enough.
Author’s response:
We thank the reviewer for the kind reminder. We have included more experimental results for assess the quality of the detection task and mood classification. Please refer to Section 4.
Reviewer’s comment 3:
It is not clear the evaluation protocol and the difficulty of the task, etc. How many different dogs (individuals) were used? How many different dog breeds were used? Were the photos acquired in the wild, etc?
Author’s response:
We thank the reviewer for the valuable comments. The training samples include the Stanford Dogs Dataset which contains at least 40 breeds and more than 500 dogs. There are various pets in training samples to enhance identification ability of faster R-CNN, ssd mobilenet and yolo identification models and avoid over-fitting. The total training images are 2761 and training step is 50000. This edit is presented as in the revised version. Please refer to Section 4.
Reviewer’s comment 4:
Is kNN suitable for real-time operation? The authors suggest new data will be used to update models, which in kNN could mean increasing the 'database of observations' and therefore increase inference time.
Author’s response:
We thank the reviewer for the kind reminder. The kNN recognition model can dump .pkl file for classification. The average time of classification in under 1 second. Therefore, the new update models will not increase inference time.
Reviewer’s comment 5:
“The total training images are 458?” is this enough? Was data augmentation used?
Author’s response:
We thank the reviewer for the Reviewer 1‘s comment about our paper. The training samples include the Stanford Dogs Dataset which contains at least 40 breeds and more than 500 dogs. The total training images are 2761 and training step is 50000.
The proposed system obtains better performance when more training images are included in training model. Please refer to Section 4.
Reviewer 2 Report
The authors successfully responded to the reviewers' comments and significantly improved the manuscript. I have no further questions.
Author Response
Revision Summary
Paper Title: Multiple Feature Dependency Detection for Deep Learning Technology – Smart Pet Surveillance System Implementation
Dear Editors and Reviewers,
We would like to express our greatest gratitude for you, the editorial team, and reviewers who have given us valuable comments to improve this paper. We have addressed all comments in the revision and listed the modifications as follows.
Ming-Fong Tsai
Department of Electronic Engineering, National United University, Taiwan.
________________________________________________________________
Response to Reviewer 2’s Comments
The authors successfully responded to the reviewers' comments and significantly improved the manuscript. I have no further questions.
Author’s response:
We thank the reviewer for the Reviewer 2‘s comment about our paper. We thanks to the reviewers and the subject editor for thoughtful critiques of our manuscript.